# An H$^1$-Galerkin Space-Time Mixed Finite Element Method for Semilinear Convection–Diffusion–Reaction Equations

Xuehui Ren [1], Siriguleng He [2] and Hong Li [1,*]

[1] School of Mathematical Sciences, Inner Mongolia University, Hohhot 010021, China; xhuiren@126.com
[2] School of Mathematical Sciences, Hohhot Minzu College, Hohhot 010021, China; cmml2005@163.com
[*] Correspondence: malhong@imu.edu.cn

**Abstract:** In this paper, the semilinear convection–diffusion–reaction equation is split into a lower-order system by introducing the auxiliary variable $q = a(x)u_x$. An H$^1$-Galerkin space-time mixed finite element method for the lower-order system is then constructed. The proposed method applies the finite element method to discretize the time and space directions simultaneously and does not require checking the Ladyzhenskaya–Babuška–Brezzi (LBB) compatibility constraints, which differs from the traditional mixed finite element method. The uniqueness of the approximate solutions $u$ and $q$ are proven. The $L^2(L^2)$ norm optimal order error estimates of the approximate solution $u$ and $q$ are derived by introducing the space-time projection operator. The numerical experiment is presented to verify the theoretical results. Furthermore, by comparing with the classical H$^1$-Galerkin mixed finite element scheme, the proposed scheme can easily improve computational accuracy and time convergence order by changing the basis function.

**Keywords:** semilinear convection–diffusion–reaction equation; H$^1$-Galerkin space-time mixed finite element method; space-time projection operator; error estimates

## 1. Introduction

The H$^1$-Galerkin space-time mixed finite element method is investigated for the one-dimensional semilinear convection–diffusion–reaction problem. The initial boundary value problem considered in this article is as follows.

Find $u = u(x,t)$ such that

$$\begin{cases} u_t - (a(x)u_x)_x + b(x)u_x + c(x)u = f(u), & (x,t) \in I \times J, \\ u(0,t) = u_t(0,t) = u(1,t) = u_t(1,t) = 0, & t \in J, \\ u(x,0) = u^0(x), & x \in I, \end{cases} \quad (1)$$

where $I = [0,1]$, and $J = (0,T]$, with $T < \infty$. The functions $a(x)$, $b(x)$, and $c(x)$ are smooth. And $0 < a_0 < a(x) < a^*$ in which $a_0$ and $a^*$ are positive constants. $u^0(x)$ is a known initial value function. We suppose that the nonlinear function $f(u)$ satisfies $f(0) = 0$, and there exists a positive constant $C$, such that

$$| f(u) | < C | u | . \quad (2)$$

Additionally, we presume that the Lipschitz condition is satisfied by the nonlinear function $f(u)$, that is, there exists a Lipschitz constant $L$, such that

$$| f(u) - f(\tilde{u}) | < L | u - \tilde{u} | . \quad (3)$$

As we know, convection–diffusion–reaction equations play an important role in describing mass and heat transport processes and reflect a wide range of physical phenomena.

It is widely used in many practical fields, for example, environmental science, electronic science, energy development, hydrology, fluid dynamics, chemistry, biology, etc. [1,2]. This kind of equation is formed when chemical reactions occur within a fluid flow. Nonlinear equations usually cannot be solved precisely; hence, it is necessary to establish and research numerical methods for an approximate solution. This encourages us to build and study efficient numerical methods for this category of nonlinear PDEs. These include but are not limited to the finite difference method [3], the adaptive finite volume element method [4], the adaptive iterative splitting method [5], and so on. When the diffusion parameter is very small, however, the numerical solutions produced by these approaches are insufficient and show non-physical oscillations. Due to this, numerous stabilized finite element methods were developed, for instance, the stabilized finite element method [6], discontinuous Galerkin time stepping with local projection stabilization [7], weak Galerkin flux-based mixed finite element method [8], semi-Lagrangian discontinuous Galerkin($DG$)-local DG method [9], space-time ultra-weak discontinuous Galerkin method [10], weak Galerkin finite element method [11], modified finite volume method [12], least-squares mixed element method [13], and the Galerkin/Least-Squares method [14], and so on.

In 1998, Pani [15] presented a type of $H^1$-Galerkin mixed finite element approach. In essential ways, this strategy differs from standard mixed methods. First, the $H^1$-Galerkin mixed finite element gains the selection range of finite element space and no longer requires that the finite element space has at least $C^1$ continuity. Second, the LBB consistency condition is ignored. Third, approximation finite element spaces $V_h$ and $W_h$ may have various polynomial degrees. Lastly, the mesh generation of finite elements does not require regularity requirements. Due to its multiple benefits, this method has been used to obtain numerical solutions to a variety of problems. In Ref. [16], Manickam investigated the semilinear reaction–diffusion problem using a higher-order exclusively discrete scheme paired with the $H^1$-Galerkin mixed finite element method. A priori error estimates for the semidiscrete scheme were studied. An implicit Runge–Kutta method was used for the temporal direction for full discretization, and the error estimates for both components were addressed. Furthermore, the $H^1$-Galerkin mixed finite element method was used to study the second-order hyperbolic equations [17], the heat conduction problem [18], the nonlinear Sine-Gordon equations [19], the regularized long wave equation [20], the second-order elliptic equations [21], two-dimensional time fractional diffusion equations [22], the Sobolev equations [23], the nonlinear Sobolev equation [24], and so on.

In these studies, the $H^1$-Galerkin mixed finite element method was used for the discretization of the space variable, and the Euler, Crank–Nicolson, or Runge–Kutta difference method was used in the time discrete formula. One of the disadvantages of these kinds of approximate schemes is that high-order accuracy in time cannot be obtained, which results in the mismatch of the convergence order between time and space discrete. A type of space-time finite element method is presented to overcome these defects. In the space-time scheme, the finite element method is employed in both time and space discretize and the high-order precision for space and time can be obtained simultaneously. The space-time method has been used to solve some time-dependent issues, for example, the traditional continuous space-time finite element method for the heat equation [25], the reduced-order method combined with the space-time finite element method for the 2D Sobolev equation [26], a high-order space-time ultra-weak discontinuous Galerkin method for the second-order wave equation [10], an $H^1$ discontinuous space-time finite element method for convection–diffusion equations [27], and so on. In these studies, the analysis technique of interpolation polynomials is introduced to prove the space-time error estimate. Here, we will perform this by introducing a space-time projection operator which is different from the above methods.

In this paper, we first obtain a coupled system equivalent to problem (1) by introducing the auxiliary variable $q = a(x)u_x$. The $H^1$-Galerkin space-time mixed finite element method is established for the coupled system (4). The mixed finite element method is extended to the space-time method here. It is a new try for a semilinear convection–diffusion–

reaction problem solving by a kind of method combining the H$^1$-Galerkin mixed method with a space-time finite element scheme. The analysis technique in this paper is different from those studies in which the interpolation technique is utilized to obtain the error of the corresponding unknown function [10,27]. The uniqueness of the approximate solutions u and q are proven. We obtain the $L^2(L^2)$ norm optimal order error estimates by introducing the space-time projections and proving the properties of these operators. The numerical example is given to demonstrate the effectiveness of the algorithm as well as the reasonableness of the theoretical analysis conclusions. Furthermore, by comparing with the classical H$^1$-Galerkin mixed finite element scheme, the proposed scheme can easily improve computational accuracy and time convergence order by changing the basis function.

The structure of this paper is as follows. The research state of the space-time finite element method and the H$^1$-Galerkin mixed finite element method, as well as the primary content of this study, are discussed in Section 1. In Section 2, we give some definitions of Sobolev spaces and the corresponding norms required for theoretical analysis in this paper. The H$^1$-Galerkin space-time mixed finite element scheme of the semilinear convection–diffusion–reaction equation is given and the uniqueness of the approximate solutions $u$ and $q$ are demonstrated. In Section 3, the $L^2(L^2)$ norm optimal order error estimates of the finite element solutions $u$ and $q$ are provided. In Section 4, a numerical example is given to verify the validity and feasibility of the scheme. Finally, some conclusions are made in Section 5.

## 2. H$^1$-Galerkin Space-Time Mixed Finite Element Scheme

Here, we will go over some fundamental concepts and definitions to understand the H$^1$-Galerkin space-time mixed finite element method of Equation (1) as well as the theoretical analysis of the numerical solutions. All of the Sobolev spaces and norms used in this paper are customary [28,29].

We will use the classical Sobolev spaces $W^{s,p}(\Omega)$, $(1 \le p \le \infty)$, as

$$W^{s,p}(\Omega) = \{u \in L^p(\Omega) \,|\, D^\alpha u \in L^p(\Omega), 0 \le |\alpha| \le s\},$$

where $L^p(\Omega) = \{u; \int_\Omega |u|^p \, dx < \infty\}$. The corresponding norm is given by

$$\|u\|_{s,p} = \{ \sum_{0 \le |\alpha| \le s} \int_\Omega (\frac{d^\alpha}{dx^\alpha} u(\cdot))^p dx \}^{\frac{1}{p}}.$$

when $p = 2$, let $W^{s,2}(\Omega) = H^s(\Omega)$. $H^s(\Omega)$ is called the $L^2(\Omega)$ space, when $s = 0$. The corresponding inner product $(\cdot, \cdot)$ and norm $\| \cdot \|$ are defined as

$$(u, v) = \int_\Omega u \cdot v dx,$$

and

$$\|u\| = [\int_\Omega |u(\cdot)|^2 \, dx]^{\frac{1}{2}}.$$

The space $H_0^1(\Omega)$ is defined as

$$H_0^1(\Omega) = \{v \in H^1(\Omega); v|_{\partial\Omega} = 0\}.$$

In addition, we also need to introduce the following space-time Sobolev spaces.

$$H^m(J; H^s(\Omega)) = \left\{ v(x,t); \int_0^T \sum_{i=0}^m \|\frac{d^i}{dt^i} v(\cdot, t)\|_s^2 dt < \infty \right\},$$

the corresponding norm $\|v\|_{H^m(J;H^s(\Omega))}$ is defined as

$$\|v\|_{H^m(J;H^s(\Omega))} = \left[ \int_0^T \sum_{i=0}^m \|\frac{d^i}{dt^i}v(\cdot,t)\|_s^2 dt \right]^{\frac{1}{2}}.$$

In particular, when $m = 0, s = 0, 1$, the corresponding norms are recorded as

$$\|v\|_{L^2(J;L^2(\Omega))} = \left[ \int_0^T \|v(\cdot,t)\|^2 dt \right]^{\frac{1}{2}},$$

and

$$\|v\|_{L^2(J;H^1(\Omega))} = \left[ \int_0^T \|v(\cdot,t)\|_1^2 dt \right]^{\frac{1}{2}}.$$

To establish the $H^1$-Galerkin space-time mixed finite element method for the problem (1), we first discretize the space and time domain $I \times \bar{J}$. Let $0 = t_0 < t_1 < \cdots < t_{N-1} = T$, subdividing the time interval $\bar{J} = [0, T]$ into the subintervals $J_n = [t_n, t_{n+1}]$, $n = 0, 1, \cdots, N - 1$. This division is represented as $\Gamma_k$, and the division unit is denoted as $Q$. The time step is

$$k_n = t_{n+1} - t_n, \quad and \quad k = \max_{0 \leq n \leq N-1} k_n, \quad n = 0, 1, \cdots, N - 1.$$

Further, we subdivide the space interval $I = [0, 1]$ into the subintervals $I_m = [x_m, x_{m+1}]$, $m = 0, 1, \cdots, M - 1$. This division is written as $\Im_h$, and the division unit is denoted by $K$. The space step is

$$h_m = x_{m+1} - x_m, \quad and \quad h = \max_{0 \leq m \leq M-1} h_m, \quad m = 0, 1, \cdots, M - 1.$$

Let $V_{hm}(I) \subset H_0^1(I)$ and $W_{hm}(I) \subset H^1(I)$, representing the space composed of piecewise continuous polynomial functions of degree $m$ defined on the subdivision $\Im_h$ for the space interval $I$, that is,

$$V_{hm} = \{v_h \in H_0^1(I); v_h|_K \in P_m(K), \quad \forall K \in \Im_h\},$$

$$W_{hm} = \{w_h \in H^1(I); w_h|_K \in P_m(K), \quad \forall K \in \Im_h\},$$

where $P_m(K)$ denotes the polynomial space defined on $K$, that degree $\leq m$.

Let $V_{kl}([0, T])$ and $W_{kl}([0, T])$, denoting the space composed of piecewise continuous polynomials functions of degree $l$ defined on the subdivision $\Gamma_k$ for the time interval $\bar{J}$, that is,

$$V_{kl}([0, T]) = \{v_k \in L^2(J); v_k|_Q \in P_l(Q), \quad \forall Q \in \Gamma_k\},$$

$$W_{kl}([0, T]) = \{w_k \in H^1(J); w_k|_Q \in P_l(Q), \quad \forall Q \in \Gamma_k\},$$

where $P_l(Q)$ denotes the polynomial space defined on $Q$, whose degree $\leq l$.

Define the space

$$V_{hk} = V_{hm} \otimes V_{kl}([0, T]),$$

$$W_{hk} = W_{hm} \otimes W_{kl}([0, T]).$$

Let $S_h^n = [0, 1] \times J_n$, which is known as the space-time slab. $V_{kl}^n$ and $W_{kl}^n$ represent the piecewise polynomial space of $V_{kl}$ and $W_{kl}$, respectively, defined on the space-time slab $S_h^n$. On this basis, $V_{hk}^n$ and $W_{hk}^n$ denote the space-time approximation polynomial space on each space-time slab $S_h^n$, that is,

$$V_{hk}^n = V_{hm} \otimes V_{kl}^n,$$

$$W_{hk}^n = W_{hm} \otimes W_{kl}^n.$$

To produce the $\text{H}^1$-Galerkin mixed space-time scheme, the semilinear convection–diffusion–reaction equation is split into a lower-order system by introducing the auxiliary variable $q = a(x)u_x$. Then, Equation (1) can be restated as the following equivalent first-order differential system.

Find $\{u, q\}$, $\forall (x, t) \in [0, 1] \times [0, T]$ such that

$$
\begin{cases}
\text{(a) } u_x = \alpha(x)q, \\
\text{(b) } u_t - q_x + \beta q + c(x)u = f(u),
\end{cases}
\tag{4}
$$

where $\alpha(x) = \frac{1}{a(x)}$, $\beta(x) = \alpha(x)b(x)$.

Let $v \in L^2(0, T; H_0^1(I))$ and $\omega \in H^1(0, T; H^1(I))$. Multiplying the formula (a) in Equation (4) by $v_x$, and the formula (b) in Equation (4) by $\omega_x$, we obtain the following weak form:

$$
\begin{cases}
\text{(a) } \displaystyle\int_0^T (u_x, v_x)\mathrm{d}t = \int_0^T (\alpha q, v_x)\mathrm{d}t, \quad \forall v \in L^2(0, T; H_0^1(I)), \\
\text{(b) } \displaystyle\int_0^T (u_t, \omega_x)\mathrm{d}t - \int_0^T (q_x, \omega_x)\mathrm{d}t + \int_0^T (\beta q, \omega_x)\mathrm{d}t + \int_0^T (c(x)u, \omega_x)\mathrm{d}t \\
\qquad\qquad = \displaystyle\int_0^T (f(u), \omega_x)\mathrm{d}t, \quad \forall \omega \in H^1(0, T; H^1(I)).
\end{cases}
\tag{5}
$$

For $\int_0^T (u_t, \omega_x)\mathrm{d}t$ in the formula (b) of Equation (5), using integration by parts, and the Dirichlet boundary condition $u_t(0, t) = u_t(1, t) = 0$, we have

$$
\int_0^T (u_t, \omega_x)\mathrm{d}t = \int_0^T [(u_t, \omega)|_0^1 - (u_{tx}, \omega)]\mathrm{d}t
$$
$$
= -\int_0^T (u_{tx}, \omega)\mathrm{d}t = -\int_0^T (\alpha q_t, \omega)\mathrm{d}t.
\tag{6}
$$

Similarly, for $\int_0^T (c(x)u, \omega_x)\mathrm{d}t$ in the formula (b) of Equation (5), using integration by parts, and the Dirichlet boundary condition $u(0, t) = u(1, t) = 0$, we have

$$
\int_0^T (c(x)u, \omega_x)\mathrm{d}t = \int_0^T [(c(x)u, \omega)|_0^1 - ((c(x)u)_x, \omega)]\mathrm{d}t
$$
$$
= -\int_0^T ((c(x)u)_x, \omega)\mathrm{d}t = -\int_0^T (\mu u, \omega)\mathrm{d}t - \int_0^T (\gamma q, \omega)\mathrm{d}t,
\tag{7}
$$

where $\mu(x) = c(x)_x$, $\gamma(x) = c(x)\alpha(x)$.

Bringing Equations (6) and (7) into the formula (b) of Equation (5), we can obtain

$$
\begin{cases}
\text{(a) } \displaystyle\int_0^T (u_x, v_x)\mathrm{d}t = \int_0^T (\alpha q, v_x)\mathrm{d}t, \quad \forall v \in L^2(0, T; H_0^1(I)), \\
\text{(b) } \displaystyle\int_0^T (\alpha q_t, \omega)\mathrm{d}t + \int_0^T (q_x, \omega_x)\mathrm{d}t = \int_0^T (\beta q, \omega_x)\mathrm{d}t - \int_0^T (\mu u, \omega)\mathrm{d}t \\
\qquad\qquad - \displaystyle\int_0^T (\gamma q, \omega)\mathrm{d}t - \int_0^T (f(u), \omega_x)\mathrm{d}t, \quad \forall \omega \in H^1(0, T; H^1(I)).
\end{cases}
\tag{8}
$$

Furthermore, when $t \in [0, T]$, we can reformulate Equation (8) as follows.

Find $\{u, q\} : [0, T] \mapsto L^2(0, T; H_0^1(I)) \times H^1(0, T; H^1(I))$, such that

$$
\begin{cases}
\text{(a) } \displaystyle\int_0^t (u_x, v_x)\mathrm{d}t = \int_0^t (\alpha q, v_x)\mathrm{d}t, \quad \forall v \in L^2(0, T; H_0^1(I)), \\
\text{(b) } \displaystyle\int_0^t (\alpha q_t, \omega)\mathrm{d}t + \int_0^t (q_x, \omega_x)\mathrm{d}t = \int_0^t (\beta q, \omega_x)\mathrm{d}t - \int_0^t (\mu u, \omega)\mathrm{d}t \\
\qquad\qquad - \displaystyle\int_0^t (\gamma q, \omega)\mathrm{d}t - \int_0^t (f(u), \omega_x)\mathrm{d}t, \quad \forall \omega \in H^1(0, T; H^1(I)).
\end{cases}
\tag{9}
$$

As a consequence, the semidiscrete $H^1$-Galerkin space-time mixed finite element scheme for Equation (9) can be expressed as follows.

Find $\{u^{hk}, q^{hk}\}$: $[t_n, t_{n+1}] \mapsto V_{hk}^n \times W_{hk}^n$, such that

$$
\begin{cases}
\text{(a)} \ \int_{t_n}^{t_{n+1}} (u_x^{hk}, v_x^{hk}) \mathrm{d}t = \int_{t_n}^{t_{n+1}} (\alpha q^{hk}, v_x^{hk}) \mathrm{d}t, \ \ \forall v^{hk} \in V_{hk}^n, \\
\text{(b)} \ \int_{t_n}^{t_{n+1}} (\alpha q_t^{hk}, \omega^{hk}) \mathrm{d}t + \int_{t_n}^{t_{n+1}} (q_x^{hk}, \omega_x^{hk}) \mathrm{d}t = \int_{t_n}^{t_{n+1}} (\beta q^{hk}, \omega_x^{hk}) \mathrm{d}t - \\
\quad \int_{t_n}^{t_{n+1}} (\mu u^{hk}, \omega^{hk}) \mathrm{d}t - \int_{t_n}^{t_{n+1}} (\gamma q^{hk}, \omega^{hk}) \mathrm{d}t - \int_{t_n}^{t_{n+1}} (f(u^{hk}), \omega_x^{hk}) \mathrm{d}t, \\
\qquad\qquad\qquad\qquad\qquad\qquad\qquad\qquad\qquad\qquad \forall \omega^{hk} \in W_{hk}^n.
\end{cases}
\tag{10}
$$

Summing Equation (10) from 1 to $N$, we can obtain

$$
\begin{cases}
\text{(a)} \ \int_0^T (u_x^{hk}, v_x^{hk}) \mathrm{d}t = \int_0^T (\alpha q^{hk}, v_x^{hk}) \mathrm{d}t, \ \ \forall v^{hk} \in V_{hk}, \\
\text{(b)} \ \int_0^T (\alpha q_t^{hk}, \omega^{hk}) \mathrm{d}t + \int_0^T (q_x^{hk}, \omega_x^{hk}) \mathrm{d}t = \int_0^T (\beta q^{hk}, \omega_x^{hk}) \mathrm{d}t - \int_0^T (\mu u^{hk}, \omega^{hk}) \mathrm{d}t \\
\qquad\qquad - \int_0^T (\gamma q^{hk}, \omega^{hk}) \mathrm{d}t - \int_0^T (f(u^{hk}), \omega_x^{hk}) \mathrm{d}t, \ \ \forall \omega^{hk} \in W_{hk}.
\end{cases}
\tag{11}
$$

when $t \in [0, T]$, we have

$$
\begin{cases}
\text{(a)} \ \int_0^t (u_x^{hk}, v_x^{hk}) \mathrm{d}t = \int_0^t (\alpha q^{hk}, v_x^{hk}) \mathrm{d}t, \ \ \forall v^{hk} \in V_{hk}, \\
\text{(b)} \ \int_0^t (\alpha q_t^{hk}, \omega^{hk}) \mathrm{d}t + \int_0^t (q_x^{hk}, \omega_x^{hk}) \mathrm{d}t = \int_0^t (\beta q^{hk}, \omega_x^{hk}) \mathrm{d}t - \int_0^t (\mu u^{hk}, \omega^{hk}) \mathrm{d}t \\
\qquad\qquad - \int_0^t (\gamma q^{hk}, \omega^{hk}) \mathrm{d}t - \int_0^t (f(u^{hk}), \omega_x^{hk}) \mathrm{d}t, \ \ \forall \omega^{hk} \in W_{hk}.
\end{cases}
\tag{12}
$$

**Theorem 1.** *Equation* (12) *has a unique solution.*

**Proof of Theorem 1.** Assuming that $(\widetilde{u}^{hk}, \widetilde{q}^{hk})$ is another solution of Equation (12), we obtain

$$
\begin{cases}
\text{(a)} \ \int_0^t ((u^{hk} - \widetilde{u}^{hk})_x, v_x^{hk}) \mathrm{d}s = \int_0^t (\alpha(q^{hk} - \widetilde{q}^{hk}), v_x^{hk}) \mathrm{d}s, \ \ \forall v^{hk} \in V^{hk}, \\
\text{(b)} \ \int_0^t (\alpha((q^{hk} - \widetilde{q}^{hk})_t), \omega^{hk}) \mathrm{d}s + \int_0^t ((q^{hk} - \widetilde{q}^{hk})_x, \omega_x^{hk}) \mathrm{d}s \\
\quad = \int_0^t (\beta(q^{hk} - \widetilde{q}^{hk}), \omega_x^{hk}) \mathrm{d}s - \int_0^t (\mu(u^{hk} - \widetilde{u}^{hk}), \omega^{hk}) \mathrm{d}s \\
\quad - \int_0^t (\gamma(q^{hk} - \widetilde{q}^{hk}), \omega^{hk}) \mathrm{d}s - \int_0^t ((f(u^{hk}) - f(\widetilde{u}^{hk})), \omega_x^{hk}) \mathrm{d}s, \ \ \forall \omega^{hk} \in W^{hk}.
\end{cases}
\tag{13}
$$

In the formula (a) of Equation (13), taking $v = u^{hk} - \widetilde{u}^{hk}$, we have

$$
\int_0^t ((u^{hk} - \widetilde{u}^{hk})_x, (u^{hk} - \widetilde{u}^{hk})_x) \mathrm{d}s = \int_0^t (\alpha(q^{hk} - \widetilde{q}^{hk}), (u^{hk} - \widetilde{u}^{hk})_x) \mathrm{d}s.
\tag{14}
$$

Using Cauchy's inequality, we obtain

$$
\begin{aligned}
&\int_0^t \| (u^{hk} - \widetilde{u}^{hk})_x \|^2 \, \mathrm{d}s \\
&\leq \frac{1}{2} \, | \alpha |_{max} \int_0^t \| q^{hk} - \widetilde{q}^{hk} \|^2 \, \mathrm{d}s + \frac{1}{2} \int_0^t \| (u^{hk} - \widetilde{u}^{hk})_x \|^2 \, \mathrm{d}s.
\end{aligned}
\tag{15}
$$

Furthermore, we have

$$\int_0^t \| (u^{hk} - \widetilde{u}^{hk})_x \|^2 \, ds \leq C \int_0^t \| q^{hk} - \widetilde{q}^{hk} \|^2 \, ds. \tag{16}$$

Thus, it holds that

$$\| (u^{hk} - \widetilde{u}^{hk})_x \|_{L^2(0,t;L^2(I))}^2 \leq C \| q^{hk} - \widetilde{q}^{hk} \|_{L^2(0,t;L^2(I))}^2. \tag{17}$$

In the formula (b) of Equation (13), choosing $\omega = q^{hk} - \widetilde{q}^{hk}$, then

$$\begin{aligned}
&\int_0^t (\alpha((q^{hk} - \widetilde{q}^{hk})_t, q^{hk} - \widetilde{q}^{hk}) ds + \int_0^t ((q^{hk} - \widetilde{q}^{hk})_x, (q^{hk} - \widetilde{q}^{hk})_x) ds \\
&= \int_0^t (\beta(q^{hk} - \widetilde{q}^{hk}), (q^{hk} - \widetilde{q}^{hk})_x) ds - \int_0^t (\mu(u^{hk} - \widetilde{u}^{hk}), q^{hk} - \widetilde{q}^{hk}) ds \\
&\quad - \int_0^t (\gamma(q^{hk} - \widetilde{q}^{hk}), q^{hk} - \widetilde{q^{hk}}) ds - \int_0^t ((f(u^{hk}) - f(\widetilde{u}^{hk})), (q^{hk} - \widetilde{q}^{hk})_x) ds.
\end{aligned} \tag{18}$$

Using Cauchy's inequality and Equation (3), we obtain

$$\begin{aligned}
&\frac{1}{2} \mid \alpha \mid_{min} \int_0^t \frac{d}{dt} \| q^{hk} - \widetilde{q}^{hk} \|^2 \, ds + \int_0^t \| (q^{hk} - \widetilde{q}^{hk})_x \|^2 \, ds \\
&\leq \frac{3}{4} \mid \beta \mid_{max} \int_0^t \| q^{hk} - \widetilde{q}^{hk} \|^2 \, ds + \frac{1}{3} \int_0^t \| (q^{hk} - \widetilde{q}^{hk})_x \|^2 \, ds \\
&\quad + \mid \gamma \mid_{max} \int_0^t \| q^{hk} - \widetilde{q}^{hk} \|^2 \, ds + \frac{3}{4} \mid \mu \mid_{max} \int_0^t \| u^{hk} - \widetilde{u}^{hk} \|^2 \, ds \\
&\quad + \frac{1}{3} \int_0^t \| q^{hk} - \widetilde{q}^{hk} \|^2 \, ds + \frac{3}{4} L \int_0^t \| u^{hk} - \widetilde{u}^{hk} \|^2 \, ds \\
&\quad + \frac{1}{3} \int_0^t \| (q^{hk} - \widetilde{q}^{hk})_x \|^2 \, ds.
\end{aligned} \tag{19}$$

Then

$$\begin{aligned}
&\frac{1}{2} \mid \alpha \mid_{min} \| q^{hk} - \widetilde{q}^{hk} \|^2 + \frac{1}{3} \int_0^t \| (q^{hk} - \widetilde{q}^{hk})_x \|^2 \, ds \\
&\leq \frac{3}{4} \mid \beta \mid_{max} \int_0^t \| q^{hk} - \widetilde{q}^{hk} \|^2 \, ds + \mid \gamma \mid_{max} \int_0^t \| q^{hk} - \widetilde{q}^{hk} \|^2 \, ds \\
&\quad + \frac{3}{4} \mid \mu \mid_{max} \int_0^t \| u^{hk} - \widetilde{u}^{hk} \|^2 \, ds + \frac{1}{3} \int_0^t \| q^{hk} - \widetilde{q}^{hk} \|^2 \, ds \\
&\quad + \frac{3}{4} L \int_0^t \| u^{hk} - \widetilde{u}^{hk} \|^2 \, ds.
\end{aligned} \tag{20}$$

Thus, it holds that

$$\begin{aligned}
&\| q^{hk} - \widetilde{q}^{hk} \|^2 + \int_0^t \| (q^{hk} - \widetilde{q}^{hk})_x \|^2 \, ds \\
&\leq C(\int_0^t \| q^{hk} - \widetilde{q}^{hk} \|^2 \, ds + \int_0^t \| u^{hk} - \widetilde{u}^{hk} \|^2 \, ds).
\end{aligned} \tag{21}$$

Furthermore, we obtain

$$\| q^{hk} - \widetilde{q}^{hk} \|^2 \leq C(\int_0^t \| q^{hk} - \widetilde{q}^{hk} \|^2 \, ds + \int_0^t \| u^{hk} - \widetilde{u}^{hk} \|^2 \, ds). \tag{22}$$

Owing to $\| u^{hk} - \widetilde{u}^{hk} \| \leq C \| (u^{hk} - \widetilde{u}^{hk})_x \|$, combining with Equation (16), we have

$$\| q^{hk} - \widetilde{q}^{hk} \|^2 \leq C \int_0^t \| q^{hk} - \widetilde{q}^{hk} \|^2 \, \mathrm{d}s. \tag{23}$$

According to Gronwall's Lemma, we have $\| q^{hk} - \widetilde{q}^{hk} \| \leq 0$, thus $q^{hk} = \widetilde{q}^{hk}$. Owing to $q^{hk} = \widetilde{q}^{hk}$, through Equation (16), we thus have $u^{hk} = \widetilde{u}^{hk}$. Therefore, Equation (12) has a unique solution. Then, we complete the proof. □

## 3. Error Estimations of Approximate Solution

We first give several associated space-time projections and prove their properties to analyze the error estimation of $\{u^{hk}, q^{hk}\}$. Introduce the Ritz projection $P_x^u : H_0^1(I) \to V_{hm}(I)$, which ensures that if $u \in H_0^1(I)$, $P_x^u u \in V_{hm}(I)$ satisfies

$$((P_x^u u)_x, \varphi_x) = (u_x, \varphi_x), \quad \forall \varphi \in V_{hm}(I). \tag{24}$$

In the sense of the $L^2$ inner product, the above operator can be further extended to space-time projection $P_x^u$ (for simplicity, we still denote the space-time projection as $P_x^u$, the same as below). Then $P_x^u : L^2(0, T; H_0^1(I)) \to V_{hm}(I) \times L^2(0, T)$ is defined as

$$\int_0^T ((P_x^u u)_x, \varphi_x) \mathrm{d}t = \int_0^T (u_x, \varphi_x) dt, \quad \forall \varphi \in V_{hm}(I) \times L^2(0, T). \tag{25}$$

Further, we introduce the projection $P_t^u : L^2(0, T) \to V_{kl}([0, T])$, such that if $u \in L^2(0, T)$, then $P_t^u u \in V_{kl}([0, T])$ satisfies

$$\int_0^T (P_t^u u) \delta \mathrm{d}t = \int_0^T u \delta \mathrm{d}t, \quad \forall \delta \in V_{kl}([0, T]). \tag{26}$$

Similarly, it can be further extended to space-time projection $P_t^u : L^2(0, T; H_0^1(I)) \to H_0^1(I) \times V_{kl}([0, T])$, such that

$$\int_0^T (P_t^u u, \delta) \mathrm{d}t = \int_0^T (u, \delta) \mathrm{d}t, \quad \forall \delta \in H_0^1(I) \times V_{kl}([0, T]). \tag{27}$$

**Lemma 1** ([25,28,30,31]). *Let $P_x^u$ and $P_t^u$ be defined as Equations (24)–(27), then the following conclusions can be established.*

*(1) Suppose $u \in H^2(0, T; H^2(I))$, such that*

$$(P_x^u u)_x = P_x^u u_x, \quad (P_t^u u)_t = P_t^u u_t, \quad (P_x^u u_x)_t = (P_x^u u_t)_x \quad (P_t^u u_x)_t = (P_t^u u_t)_x,$$

$$P_x^u P_t^u u = P_t^u P_x^u u, \quad \| P_x^u u_x \| \leq \| u_x \|, \quad \| P_t^u u \|_{L^2(J)} \leq \| u \|_{L^2(J)}. \tag{28}$$

*(2) Suppose $u \in H^1(0, T) \cap H^r(0, T)$, there exists a positive constant C independent of the time step k, satisfying*

$$\| P_t^u u - u \|_{H^s(0,T)} \leq Ck^{r-s} \| u \|_{H^r(0,T)}, \quad s = 0, 1, \ 1 \leq r \leq l + 1. \tag{29}$$

*(3) Suppose $u \in H_0^1(I) \cap H^r(I)$, there exists a positive constant C independent of the space step h, satisfying*

$$\| P_x^u u - u \|_s \leq Ch^{r-s} \| u \|_r, \quad s = 0, 1, \ 1 \leq r \leq m + 1. \tag{30}$$

*(4) Suppose $u \in L^2(0, T; H^r(I)) \cap H^1(0, T; H_0^1(I))$, such that*

$$\| (u - P_x^u u)(t) \|_{L^2(0,T;H^s(I))} \leq Ch^{r-s} \| u(t) \|_{L^2(0,T;H^r(I))}, \quad 1 \leq r \leq m + 1, s = 0, 1. \tag{31}$$

(5)    Suppose $u \in L^2(0,T;H_0^1(I)) \cap H^r(0,T;H_0^1(I))$, such that

$$\| (u - P_t^u u)(t) \|_{H^s(0,T;H_0^1(I))} \leq Ck^{r-s} \| u(t) \|_{H^r(0,T;H_0^1(I))}, \quad 1 \leq r \leq l+1, s = 0,1. \quad (32)$$

(6)    Suppose $u \in H^{l+1}(0,T;H^s(I)) \cap L^2(0,T;H^{m+1}(I))$, $s = 0,1$, such that

$$\| u - P_x^u P_t^u u \|_{L^2(0,T;H^s(I))} \leq C\{h^{m+1-s} \| u \|_{L^2(0,T;H^{m+1}(I))} + k^{l+1} \| u \|_{H^{l+1}(0,T;H^s(I))}\}. \quad (33)$$

**Proof of (1) in Lemma 1.** For $\forall \varphi \in V_{hk}(I) \cap L^2(0,T;H_0^1(I))$, we obtain

$$\int_0^T ((P_x^u u)_x, \varphi_x) \mathrm{d}t = \int_0^T (u_x, \varphi_x) \mathrm{d}t = -\int_0^T (u_{xx}, \varphi) \mathrm{d}t$$

$$= -\int_0^T ((P_x^u u_x)_x, \varphi) \mathrm{d}t = \int_0^T (P_x^u u_x, \varphi_x) \mathrm{d}t.$$

We can obtain the result $(P_x^u u)_x = P_x^u u_x$.

Further, let $\forall \varphi \in V_{hk}(I) \cap H^1(0,T;H_0^1(I))$ be an arbitrary function with $\varphi_x(\cdot, 0) = \varphi_x(\cdot, T) = 0$, then

$$\int_0^T ((P_x^u u_t)_x, \varphi_x) \mathrm{d}t = \int_0^T (u_{tx}, \varphi_x) \mathrm{d}t = -\int_0^T (u_x, \varphi_{tx}) \mathrm{d}t$$

$$= -\int_0^T ((P_x^u u)_x, \varphi_{tx}) \mathrm{d}t = -\int_0^T (P_x^u u_x, \varphi_{tx}) \mathrm{d}t$$

$$= \int_0^T ((P_x^u u_x)_t, \varphi_x) \mathrm{d}t.$$

Then $(P_x^u u_x)_t = (P_x^u u_t)_x$.

Let $\forall \varphi \in V_{hk}(I) \cap H^1(0,T;H_0^1(I))$ be an arbitrary function with $\varphi_x(\cdot, 0) = \varphi_x(\cdot, T) = 0$, we obtain

$$\int_0^T (P_t^u u_t, \varphi_x) \mathrm{d}t = \int_0^T (u_t, \varphi_x) \mathrm{d}t = -\int_0^T (u, \varphi_{tx}) \mathrm{d}t$$

$$= -\int_0^T (P_t^u u, \varphi_{tx}) \mathrm{d}t = \int_0^T ((P_t^u u)_t, \varphi_x) \mathrm{d}t.$$

Indeed, we obtain the result $(P_t^u u)_t = P_t^u u_t$.

Further, for $\forall \varphi \in V_{hk}(I) \cap L^2(0,T;H_0^1(I))$, we have

$$\int_0^T ((P_t^u u_x)_t, \varphi) \mathrm{d}t = \int_0^T (P_t^u u_{xt}, \varphi) \mathrm{d}t = \int_0^T (u_{xt}, \varphi) \mathrm{d}t$$

$$= -\int_0^T (u_t, \varphi_x) \mathrm{d}t = -\int_0^T (P_t^u u_t, \varphi_x) \mathrm{d}t$$

$$= \int_0^T (((P_t^u u_t)_x, \varphi) \mathrm{d}t.$$

Then $(P_t^u u_t)_x = (P_t^u u_x)_t$.

Similarly, for $\forall \varphi \in V_{hk}(I) \cap L^2(0,T;H^2(I))$, we have

$$\int_0^T (P_x^u P_t^u u, \varphi_{xx}) \mathrm{d}t = -\int_0^T ((P_x^u P_t^u u)_x, \varphi_x) \mathrm{d}t = -\int_0^T ((P_t^u u)_x, \varphi_x) \mathrm{d}t$$

$$= \int_0^T (P_t^u u, \varphi_{xx}) \mathrm{d}t = \int_0^T (u, \varphi_{xx}) \mathrm{d}t$$

$$= -\int_0^T (u_x, \varphi_x) \mathrm{d}t = -\int_0^T ((P_x^u u)_x, \varphi_x) \mathrm{d}t$$

$$= \int_0^T (P_x^u u, \varphi_{xx}) \mathrm{d}t + \int_0^T (P_t^u P_x^u u, \varphi_{xx}) \mathrm{d}t.$$

Therefore, we can obtain the result $P_x^u P_t^u u = P_t^u P_x^u u$.
For $\forall \varphi \in V_{hk}(I) \cap L^2(0,T;H_0^1(I))$, it holds that

$$(P_x^u u_x, \varphi_x) = (u_x, \varphi_x).$$

Taking $\varphi = P_x^u u$ in Equation (24), we obtain

$$(P_x^u u_x, P_x^u u_x) = (u_x, P_x^u u_x).$$

Using the Schwartz's inequality, we have

$$\| P_x^u u_x \|^2 \leq \| u_x \| \cdot \| P_x^u u_x \|.$$

Then, we can obtain $\| P_x^u u_x \| \leq \| u_x \|$.
Taking $\delta = P_t^u u$ in Equation (26), we obtain

$$\int_0^T (P_t^u u)^2 \mathrm{d}t = \int_0^T u P_t^u u \, \mathrm{d}t.$$

By Cauchy's inequality, we can obtain $\| P_t^u u \|_{L^2(J)} \leq \| u \|_{L^2(J)}$.

The conclusion of Equation (1) in Lemma 1 is proven. Combining Equations (31) and (32), we obtain the scheme (33). The remaining conclusions in Lemma 1 are the standard results of finite element analysis.   □

Further, we have

$$\| u - P_x^u P_t^u u \|_{L^2(0,T;H^s(I))} \leq \| u - P_x^u u \|_{L^2(0,T;H^s(I))} + \| P_x^u u - P_x^u P_t^u u \|_{L^2(0,T;H^s(I))}$$

$$\leq \| u - P_x^u u \|_{L^2(0,T;H^s(I))} + \| P_x^u(u - P_t^u u) \|_{L^2(0,T;H^s(I))}$$

$$\leq \| u - P_x^u u \|_{L^2(0,T;H^s(I))} + \| u - P_t^u u \|_{L^2(0,T;H^s(I))}.$$

To give an error estimate of $u^{hk}$, first of all, we give the equation that the error $u - u^{hk}$ satisfies. From the formula (a) of Equations (9) and (12), we can obtain

$$\int_0^t ((u - u^{hk})_x, v_x^{hk}) \mathrm{d}t = \int_0^t (\alpha(q - q^{hk}), v_x^{hk}) \mathrm{d}t. \tag{34}$$

The error is split as follows to determine error estimates for semidiscrete approximations $u - u^{hk} = (u - P_x^u P_t^u u) + (P_x^u P_t^u u - u^{hk}) = \rho + \theta$. Since the estimates of $\rho$ are known from Lemma 1, it is enough to estimate $\theta$. We first give the equation that $\theta$ satisfies.

**Lemma 2.** *Suppose $P_x^u$ and $P_t^u$ be defined as (24)–(27), if $u \in L^2(0,T;H_0^1(I))$, $\forall v^{hk} \in V_{hk}$, it obtains*

$$\int_0^t (\theta_x, v_x^{hk}) \mathrm{d}t = \int_0^t ((P_t^u u - u)_x, v_x^{hk}) \mathrm{d}t + \int_0^t (\alpha(q - q^{hk}), v_x^{hk}) \mathrm{d}t. \tag{35}$$

**Proof of Lemma 2.** According to Lemma 1 and Equation (34), we have

$$\int_0^t ((P_x^u P_t^u u - u^{hk})_x, v_x^{hk})dt = \int_0^t ((P_t^u u - u^{hk})_x, v_x^{hk})dt$$

$$= \int_0^t ((P_t^u u - u)_x, v_x^{hk})dt + \int_0^t ((u - u^{hk})_x, v_x^{hk})dt$$

$$= \int_0^t ((P_t^u u - u)_x, v_x^{hk})dt + \int_0^t (\alpha(q - q^{hk}), v_x^{hk})dt.$$

The conclusion of Lemma 2 is proven.  □

**Lemma 3.** *Let* $u \in L^2(0, T; H_0^1(I))$, *then we obtain an error estimate of* $\theta$

$$\int_0^t \| \theta \|^2 dt \leq C \int_0^t [\| (P_t^u u - u)_x \|^2 + \| q - q^{hk} \|^2]dt. \tag{36}$$

**Proof of Lemma 3.** Taking $v^{hk} = \theta$ in Equation (35) of Lemma 2, we obtain

$$\int_0^t (\theta_x, \theta_x)dt = \int_0^t ((P_t^u u - u)_x, \theta_x)dt + \int_0^t (\alpha(q - q^{hk}), \theta_x)dt. \tag{37}$$

Applying Hölder's inequality and Cauchy's inequality, we have

$$\int_0^t \| \theta_x \|^2 dt \leq \frac{3}{4} \int_0^t \| (P_t^u u - u)_x \|^2 dt + \frac{1}{3} \int_0^t \| \theta_x \|^2 dt$$

$$+ \frac{3}{4} \mid \alpha \mid_{max} \int_0^t \| q - q^{hk} \|^2 dt + \frac{1}{3} \int_0^t \| \theta_x \|^2 dt. \tag{38}$$

Furthermore, we obtain

$$\frac{1}{3} \int_0^t \| \theta_x \|^2 dt \leq \frac{3}{4} \int_0^t \| (P_t^u u - u)_x \|^2 dt + \frac{3}{4} \mid \alpha \mid_{max} \int_0^t \| q - q^{hk} \|^2 dt. \tag{39}$$

Owing to $\| \theta \| \leq C \| \theta_x \|$, we obtain

$$\int_0^t \| \theta \|^2 dt \leq C \int_0^t [\| (P_t^u u - u)_x \|^2 + \| q - q^{hk} \|^2]dt. \tag{40}$$

Then, we complete the proof.  □

To consider the error estimate of the intermediate variable $q^{hk}$. We introduce the Ritz projection $P_x^q : H^1(I) \rightarrow W_{hm}(I)$, such that if $q \in H^1(I)$, $P_x^q q \in W_{hm}(I)$ satisfies

$$((P_x^q q)_x, \varphi_x) = (q_x, \varphi_x), \quad \forall \varphi \in W_{hm}(I). \tag{41}$$

And the approximation properties of $P_x^q$ are satisfied

$$\| P_x^q q_x \| \leq \| q_x \|, \quad \| P_x^q q - q \|_s \leq Ch^{r-s} \| q \|_r, \quad q \in H^r(I), \quad 1 \leq r \leq m+1, s = 0.$$

In the sense of $L^2$, the above operator can be further extended to space-time projection $P_x^q : L^2(0, T; H^1(I)) \rightarrow W_{hm}(I) \times L^2(0, T)$ defined below

$$\int_0^T ((P_x^q q)_x, \varphi_x)dt = \int_0^T (q_x, \varphi_x)dt, \quad \forall \varphi \in W_{hm}(I) \times L^2(0, T). \tag{42}$$

We introduce projection $P_t^q : H^1(0,T) \to W_{kl}([0,T])$, such that if $q \in H^1(0,T)$, then $P_t^q q \in W_{kl}([0,T])$ satisfies

$$\int_0^T (P_t^q q)_t \zeta \mathrm{d}t = \int_0^T q_t \zeta \mathrm{d}t, \quad \forall \zeta \in W_{kl}([0,T]). \tag{43}$$

The following approximation properties hold.

$$\| P_t^q q \|_{H^1(J)} \leq \| q \|_{H^1(J)}, \quad \forall q \in H^r(0,T) \cap H^1(0,T),$$

$$\| P_t^q q - q \|_{H^s(0,T)} \leq Ck^{r-s} \| q \|_{H^r(0,T)}, \quad (-l+1 \leq s \leq 1 \leq r \leq l+1).$$

In the sense of $L^2$ inner product, $P_t^q$ defined in Equation (43) can be further extended to space-time projection $P_t^q : H^1(0,T;L^2(I)) \to H^1(I) \times W_{kl}([0,T])$ defined as below

$$\int_0^T ((P_t^q q)_t, \zeta) \mathrm{d}t = \int_0^T (q_t, \zeta) \mathrm{d}t, \quad \forall \zeta \in H^1(I) \times W_{kl}([0,T]). \tag{44}$$

**Lemma 4** ([25,28,30,31]). *Let $P_x^q$ and $P_t^q$ be defined as Equations (41)–(44), then the following conclusion can be established.*

*(1)    Suppose $q \in H^2(0,T;H^2(I)) \cap H^1(0,T;L^2(I))$, then*

$$(P_x^q q)_x = P_x^q q_x, \quad (P_x^q q_t)_x = (P_x^q q_x)_t, \quad (P_t^q q)_t = P_t^q q_t, \quad (P_t^q q_t)_x = (P_t^q q_x)_t,$$
$$P_x^q P_t^q q_t = P_t^q P_x^q q_t, \quad P_x^q P_t^q q_x = P_t^q P_x^q q_x, \quad P_x^q P_t^q q = P_t^q P_x^q q. \tag{45}$$

*(2)    Suppose $q \in H^1(0,T;H^1(I)) \cap H^r(0,T;H^1(I))$, then it holds that*

$$\| (q - P_t^q q)(t) \|_{H^s(0,T;H^1(I))} \leq Ck^{r-s} \| q(t) \|_{H^r(0,T;H^1(I))}, \quad s = 0,1, 0 \leq r \leq l+1. \tag{46}$$

*(3)    Suppose $q \in H^1(0,T;H^r(I)) \cap H^1(0,T;H^1(I))$, then*

$$\| (q - P_x^q q)(t) \|_{H^1(0,T;H^s(I))} \leq Ch^{r-s} \| q(t) \|_{H^1(0,T;H^r(I))}, \quad 1 \leq r \leq m+1, s = 0,1. \tag{47}$$

*(4)    Suppose $q \in H^{l+1}(0,T;H^s(I)) \cap H^1(0,T;H^{m+1}(I))$, and $\forall q \in H^1(0,T;H^1(I))$, $s = 0,1$, we have*

$$\| q - P_x^q P_t^q q \|_{L^2(0,T;H^s(I))} \leq C\{ h^{m+1-s} \| q \|_{L^2(0,T;H^{m+1}(I))} + k^{l+1} \| q \|_{H^{l+1}(0,T;H^s(I))} \}. \tag{48}$$

**Proof of (1) in Lemma 4.** Let $\forall \psi \in W_{hk}(I) \cap H^1(0,T;H^1(I))$, we obtain

$$\int_0^T ((P_x^q q)_x, \psi_x) \mathrm{d}t = \int_0^T (q_x, \psi_x) \mathrm{d}t = -\int_0^T ((q_x)_x, \psi) \mathrm{d}t$$
$$= -\int_0^T ((P_x^q q_x)_x, \psi) \mathrm{d}t = \int_0^T (P_x^q q_x, \psi_x) \mathrm{d}t.$$

Indeed, we can obtain the result $(P_x^q q)_x = P_x^q q_x$.

Further, let $\forall \psi \in W_{hk}(I) \cap H^1(0,T;H^1(I))$ be an arbitrary function with $\psi(\cdot, 0) = \psi(\cdot, T) = 0$; therefore, we have

$$\int_0^T ((P_x^q q_t)_x, \psi_x) \mathrm{d}t = \int_0^T (q_{tx}, \psi_x) \mathrm{d}t = -\int_0^T (q_x, \psi_{tx}) \mathrm{d}t$$
$$= -\int_0^T ((P_x^q q)_x, \psi_{tx}) \mathrm{d}t = \int_0^T (((P_x^q q)_x)_t, \psi_x) \mathrm{d}t.$$

Then $(P_x^q q_t)_x = (P_x^q q_x)_t$.

Let $\forall \psi \in W_{hk}(I) \cap H^1(0, T; H^1(I))$, then

$$\int_0^T ((P_t^q q)_t, \psi_{tx})dt = \int_0^T (q_t, \psi_{tx})dt = -\int_0^T ((q_t)_t, \psi_x)dt$$

$$= -\int_0^T ((P_t^q q_t)_t, \psi_x)dt = \int_0^T (P_t^q q_t, \psi_{tx})dt.$$

In fact, we can obtain the result $(P_t^q q)_t = P_t^q q_t$.
Further, let $\forall \psi \in W_{hk}(I) \cap H^1(0, T; H^1(I))$, then we have

$$\int_0^T ((P_t^q q_t)_x, \psi)dt = -\int_0^T (P_t^q q_t, \psi_x)dt = -\int_0^T (q_t, \psi_x)dt$$

$$= \int_0^T ((q_t)_x, \psi)dt = \int_0^T ((q_x)_t, \psi)dt$$

$$= \int_0^T (P_t^q (q_x)_t, \psi)dt = \int_0^T ((P_t^q q_x)_t, \psi)dt.$$

Then $(P_t^q q_t)_x = (P_t^q q_x)_t$.
Let $\forall \psi \in W_{hk}(I) \cap H^1(0, T; H^2(I))$, and we have

$$\int_0^T (P_x^q P_t^q q_t, \psi_{txx})dt = -\int_0^T ((P_x^q P_t^q q_t)_x, \psi_{tx})dt = -\int_0^T ((P_t^q q_t)_x, \psi_{tx})dt$$

$$= \int_0^T (P_x^q q_x, \psi_{txx})dt = -\int_0^T ((P_x^q q)_x, \psi_{txx})dt = \int_0^T (q_x, \psi_{txx})dt$$

$$= -\int_0^T ((q_x)_t, \psi_{xx})dt = -\int_0^T ((P_t^q q_x)_t, \psi_{xx})dt = -\int_0^T ((P_t^q q_x)_x, \psi_{tx})dt$$

$$= -\int_0^T ((P_x^q P_t^q q_x)_x, \psi_{tx})dt = \int_0^T (P_x^q P_t^q q_x, \psi_{txx})dt.$$

So that $P_x^q P_t^q q_x = P_t^q P_x^q q_x$.
Let $\forall \psi \in W_{hk}(I) \cap H^1(0, T; H^2(I))$, then we have

$$\int_0^T (P_x^q P_t^q q_t, \psi_{txx})dt = -\int_0^T ((P_x^q P_t^q q_t)_x, \psi_{tx})dt = -\int_0^T ((P_t^q q_t)_x, \psi_{tx})dt$$

$$= \int_0^T (P_t^q q_t, \psi_{txx})dt = \int_0^T ((P_t^q q)_t, \psi_{txx})dt = \int_0^T (q_t, \psi_{txx})dt$$

$$= -\int_0^T ((q_t)_x, \psi_{tx})dt = -\int_0^T ((P_x^q q_t)_x, \psi_{tx})dt = -\int_0^T ((P_x^q q_t)_t, \psi_{xx})dt$$

$$= -\int_0^T ((P_t^q P_x^q q_t)_t, \psi_{xx})dt = \int_0^T (P_t^q P_x^q q_t, \psi_{txx})dt.$$

So that $P_x^q P_t^q q_t = P_t^q P_x^q q_t$.

Let $\forall \psi \in W_{hk}(I) \cap H^1(0, T; H^2(I))$, then we have

$$\int_0^T (P_x^q P_t^q q, \psi_{txx})\mathrm{d}t = -\int_0^T ((P_x^q P_t^q q)_x, \psi_{tx})\mathrm{d}t = -\int_0^T ((P_t^q q)_x, \psi_{tx})\mathrm{d}t$$

$$= -\int_0^T (P_t^q q_x, \psi_{tx})\mathrm{d}t = \int_0^T ((P_t^q q_x)_t, \psi_x)\mathrm{d}t = \int_0^T ((q_x)_t, \psi_x)\mathrm{d}t$$

$$= -\int_0^T (q_x, \psi_{tx})\mathrm{d} = -\int_0^T ((P_x^q q)_x, \psi_{tx})\mathrm{d}t = \int_0^T (P_x^q q, \psi_{txx})\mathrm{d}t$$

$$= -\int_0^T ((P_x^q q)_t, \psi_{xx})\mathrm{d}t = -\int_0^T ((P_t^q P_x^q q)_t, \psi_{xx})\mathrm{d}t = \int_0^T (P_t^q P_x^q q, \psi_{txx})\mathrm{d}t.$$

The conclusion of Equation (1) in Lemma 4 is proven. The remaining conclusions in Lemma 4 are the standard results of finite element analysis. □

To prove an error estimation of $q^{hk}$, we first present the error equation $q - q^{hk}$ satisfied. By the formula (b) of Equations (9) and (12), we obtain

$$\int_0^t (\alpha(q - q^{hk})_t, \omega^{hk})\mathrm{d}t + \int_0^t ((q - q^{hk})_x, \omega_x^{hk})\mathrm{d}t$$

$$= \int_0^t (\beta(q - q^{hk}), \omega_x^{hk})\mathrm{d}t - \int_0^t (\mu(u - u^{hk}), \omega^{hk})\mathrm{d}t \tag{49}$$

$$- \int_0^t (\gamma(q - q^{hk}), \omega^{hk})\mathrm{d}t - \int_0^t ((f(u) - f(u^{hk})), \omega_x^{hk})\mathrm{d}t.$$

The error split as follows to determine error estimates for semidiscrete approximations $q - q^{hk} = (q - P_t^q P_x^q q) + (P_t^q P_x^q q - q^{hk}) = \eta + \xi$. From Lemma 4, it is sufficient to estimate $\xi$ because the estimations of $\eta$ are known. To analyze $\xi$, we first give the equation that $\xi$ satisfies.

**Lemma 5.** $P_x^q$ and $P_t^q$ defined as Equations (41)–(44), for $\forall \omega^{hk} \in W_{hk}$, $q \in H^1(0, T; H^1(I))$, then

$$\int_0^t (\alpha \xi_t, \omega^{hk})\mathrm{d}t + \int_0^t (\xi_x, \omega_x^{hk})\mathrm{d}t = \int_0^t ((P_x^q q - q)_t, \omega^{hk})\mathrm{d}t$$

$$+ \int_0^t [((P_t^q q - q)_x, \omega_x^{hk}) + ((1 - \alpha)\eta_t, \omega^{hk}) + (\beta(q - q^{hk}), \omega_x^{hk})]\mathrm{d}t \tag{50}$$

$$- \int_0^t [(\mu(u - u^{hk}), \omega^{hk}) + (\gamma(q - q^{hk}), \omega^{hk}) + ((f(u) - f(u^{hk})), \omega_x^{hk})]\mathrm{d}t.$$

**Proof of Lemma 5.** According to Lemma 4 and Equation (49), we obtain

$$\int_0^t ((P_t^q P_x^q q - q^{hk})_t, \omega^{hk})\mathrm{d}t + \int_0^t ((P_t^q P_x^q q - q^{hk})_x, \omega_x^{hk})\mathrm{d}t$$

$$= \int_0^t [((P_x^q q - q^{hk})_t, \omega^{hk}) + ((P_t^q q - q^{hk})_x, \omega_x^{hk})]\mathrm{d}t = \int_0^t [((P_x^q q - q)_t, \omega^{hk})$$

$$+ ((P_t^q q - q)_x, \omega_x^{hk}) + ((q - q^{hk})_t, \omega^{hk}) + ((q - q^{hk})_x, \omega_x^{hk}))]\mathrm{d}t$$

$$= \int_0^t [((P_x^q q - q)_t, \omega^{hk}) + ((P_t^q q - q)_x, \omega_x^{hk}) + ((1 - \alpha)\eta_t, \omega^{hk}) \tag{51}$$

$$+ (\beta(q - q^{hk}), \omega_x^{hk})]\mathrm{d}t - \int_0^t [(\mu(u - u^{hk}), \omega^{hk}) + \gamma(q - q^{hk}), \omega^{hk})$$

$$+ (((f(u) - f(u^{hk})), \omega_x^{hk})]\mathrm{d}t + \int_0^t ((1 - \alpha)(P_t^q P_x^q q - q^{hk})_t, \omega^{hk})\mathrm{d}t.$$

Furthermore, we have

$$\int_0^t (\alpha \xi_t, \omega^{hk}) \mathrm{d}t + \int_0^t (\xi_x, \omega_x^{hk}) \mathrm{d}t = \int_0^t ((P_x^q q - q)_t, \omega^{hk}) \mathrm{d}t$$

$$+ \int_0^t [((P_t^q q - q)_x, \omega_x^{hk}) + ((1 - \alpha)\eta_t, \omega^{hk}) + (\beta(q - q^{hk}), \omega_x^{hk})] \mathrm{d}t \tag{52}$$

$$- \int_0^t [(\mu(u - u^{hk}), \omega^{hk}) + (\gamma(q - q^{hk}), \omega^{hk}) + ((f(u) - f(u^{hk})), \omega_x^{hk})] \mathrm{d}t.$$

The proof is then completed. □

**Theorem 2.** *Let $q$ and $q^{hk}$ be the solutions of Equations (9) and (12), respectively. For $\forall t \in [0, T]$, $q \in H^{l+1}(0, T; H^1(I)) \cap L^2(0, T; H^{m+1}(I))$, then the following estimation holds.*

$$\| \xi \|_{L^2(0,T;L^2(I))} \leq C \{ h^{m+1} [\| q \|_{H^1(0,T;H^{m+1}(I))} + \| u \|_{L^2(0,T;H^{m+1}(I))}]$$

$$+ k^{l+1} [\| q \|_{H^{l+1}(0,T;H^1(I))} + \| q_t \|_{H^{l+1}(0,T;L^2(I))} + \| u \|_{H^{l+1}(0,T;L^2(I))}] \}. \tag{53}$$

**Proof of Theorem 2.** Selecting $\omega^{hk} = \xi$ in Equation (50) of Lemma 5, we obtain

$$\int_0^t (\alpha \xi_t, \xi) \mathrm{d}t + \int_0^t (\xi_x, \xi_x) \mathrm{d}t = \int_0^t ((P_x^q q - q)_t, \xi) \mathrm{d}t$$

$$+ \int_0^t [((P_t^q q - q)_x, \xi_x) + ((1 - \alpha)\eta_t, \xi) + (\beta(q - q^{hk}), \xi_x)] \mathrm{d}t \tag{54}$$

$$- \int_0^t [(\mu(u - u^{hk}), \xi) + (\gamma(q - q^{hk}), \xi) + ((f(u) - f(u^{hk})), \xi_x)] \mathrm{d}t.$$

Applying Hölder's inequality and Cauchy's inequality, and noticing the Lipschitz conditions $f$ satisfied, yields

$$\frac{1}{2} \mid \alpha \mid_{min} \int_0^t \frac{d}{dt} \| \xi \|^2 \, \mathrm{d}t + \int_0^t \| \xi_x \|^2 \, \mathrm{d}t$$

$$\leq \int_0^t [\| (P_x^q q - q)_t \|^2 + \frac{1}{4} \| \xi \|^2 + \| (P_t^q q - q)_x \|^2 + \frac{1}{4} \| \xi_x \|^2] \mathrm{d}t$$

$$+ \int_0^t [\| (1 - \alpha)\eta_t \|^2 + \frac{1}{4} \| \xi \|^2 + \| \beta(\eta + \xi) \|^2 + \frac{1}{4} \| \xi_x \|^2 + \| \gamma(\eta + \xi) \|^2] \mathrm{d}t \tag{55}$$

$$+ \int_0^t [\frac{1}{4} \| \xi \|^2 + \| \mu(\rho + \theta) \|^2 + \frac{1}{4} \| \xi \|^2 + \| L(\rho + \theta) \|^2 + \frac{1}{4} \| \xi_x \|^2] \mathrm{d}t.$$

Further, we obtain

$$\frac{1}{2} \mid \alpha \mid_{min} \| \xi \|^2 + \frac{1}{4} \int_0^t \| \xi_x \|^2 \, \mathrm{d}t$$

$$\leq \int_0^t [\| (P_x^q q - q)_t \|^2 + \| (P_t^q q - q)_x \|^2 + \| (1 - \alpha)\eta_t \|^2 + \| \beta(\eta + \xi) \|^2] \mathrm{d}t \tag{56}$$

$$+ \int_0^t [\| \gamma(\eta + \xi) \|^2 + \| \mu(\rho + \theta) \|^2 + \| L(\rho + \theta) \|^2] \mathrm{d}t + \int_0^t \| \xi \|^2 \, \mathrm{d}t.$$

Using the triangle inequality, we find that

$$\| \xi \|^2 \le C \int_0^t [\| (P_x^q q - q)_t \|^2 + \| (P_t^q q - q)_x \|^2 + \| \eta \|^2] \mathrm{d}t$$
$$+ \int_0^t [\| \eta_t \|^2 + \| \rho \|^2 + \| \theta \|^2] \mathrm{d}t + C \int_0^t \| \xi \|^2 \, \mathrm{d}t. \tag{57}$$

From Lemma 3, we have

$$\| \xi \|^2 \le C \int_0^t [\| (P_x^q q - q)_t \|^2 + \| (P_t^q q - q)_x \|^2 + \| (P_t^u u - u)_x \|^2] \mathrm{d}t$$
$$+ \int_0^t [\| \eta \|^2 + \| \eta_t \|^2 + \| \rho \|^2] \mathrm{d}t + C \int_0^t \| \xi \|^2 \, \mathrm{d}t. \tag{58}$$

Using the Gronwall's Lemma, we have

$$\| \xi \|^2 \le C \int_0^t [\| (P_x^q q - q)_t \|^2 + \| (P_t^q q - q)_x \|^2 + \| (P_t^u u - u)_x \|^2] \mathrm{d}t$$
$$+ \int_0^t [\| \eta \|^2 + \| \eta_t \|^2 + \| \rho \|^2] \mathrm{d}t. \tag{59}$$

Combining Lemmas 1 and 4, we obtain

$$\| \xi \|_{L^2(0,T;L^2(I))} \le C \{ h^{m+1} [\| q \|_{H^1(0,T;H^{m+1}(I))} + \| u \|_{L^2(0,T;H^{m+1}(I))}]$$
$$+ k^{l+1} [\| q \|_{H^{l+1}(0,T;H^1(I))} + \| q_t \|_{H^{l+1}(0,T;L^2(I))} + \| u \|_{H^{l+1}(0,T;L^2(I))}] \}. \tag{60}$$

The conclusion in Theorem 2 is proven.  □

Based on Theorem 2, with the error estimate of $\xi$, and $\theta$ satisfying Equation (36), the following theorem can be obtained.

**Theorem 3.** *Let $u$ and $u^{hk}$ be the solutions of Equations (9) and (12), respectively. Then $\forall u \in H^{l+1}(0,T;L^2(I)) \cap L^2(0,T;H^{m+1}(I))$, and $\forall t \in [0,T]$, we can obtain*

$$\| \theta \|_{L^2(0,T;L^2(I))} \le C \{ h^{m+1} [\| q \|_{H^1(0,T;H^{m+1}(I))} + \| u \|_{L^2(0,T;H^{m+1}(I))}]$$
$$+ k^{l+1} [\| q \|_{H^{l+1}(0,T;H^1(I))} + \| q_t \|_{H^{l+1}(0,T;L^2(I))} + \| u \|_{H^{l+1}(0,T;L^2(I))}] \}. \tag{61}$$

In Lemma 3, we have already discussed the estimation of $\theta$, so we only need to substitute Equation (53) into Equation (36), and we can obtain Theorem 3.

Through the conclusions of Theorem 2, Lemma 4, Theorem 3, and Lemma 1, we can obtain the $L^2(L^2)$ norm error estimate of $q - q^{hk}$ and $u - u^{hk}$.

**Theorem 4.** *Let $q$, $u$, and $q^{hk}$, $u^{hk}$ be the solutions of Equations (9) and (12), respectively. $\forall q \in H^{l+1}(0,T;H^1(I)) \cap L^2(0,T;H^{m+1}(I)), \forall u \in H^{l+1}(0,T;L^2(I)) \cap L^2(0,T;H^{m+1}(I))$, and $\forall t \in [0,T]$. Then we obtain an error estimate of $\| q - q^{hk} \|_{L^2(0,T;L^2(I))}$, $\| u - u^{hk} \|_{L^2(0,T;L^2(I))}$*

$$\| q - q^{hk} \|_{L^2(0,T;L^2(I))} \le C \{ h^{m+1} [\| q \|_{H^1(0,T;H^{m+1}(I))} + \| u \|_{L^2(0,T;H^{m+1}(I))}]$$
$$+ k^{l+1} [\| q \|_{H^{l+1}(0,T;H^1(I))} + \| q_t \|_{H^{l+1}(0,T;L^2(I))} + \| u \|_{H^{l+1}(0,T;L^2(I))}] \}, \tag{62}$$

*and*

$$\| u - u^{hk} \|_{L^2(0,T;L^2(I))} \le C \{ h^{m+1} [\| q \|_{H^1(0,T;H^{m+1}(I))} + \| u \|_{L^2(0,T;H^{m+1}(I))}]$$
$$+ k^{l+1} [\| q \|_{H^{l+1}(0,T;H^1(I))} + \| q_t \|_{H^{l+1}(0,T;L^2(I))} + \| u \|_{H^{l+1}(0,T;L^2(I))}] \}. \tag{63}$$

Based on the relative error estimation results of $u - u^{hk}$ and $q - q^{hk}$, when $t = T$, we have an overall error estimate conclusion.

**Corollary 1.** *The following estimation formulas are established with the assumptions of Theorems 2 and 3.*

$$
\begin{aligned}
\| q - q^{hk} \|_{L^2(0,T;L^2(I))} &\leq C\{h^{m+1}[\| q \|_{H^1(0,T;H^{m+1}(I))} + \| u \|_{L^2(0,T;H^{m+1}(I))}] \\
&+ k^{l+1}[\| q \|_{H^{l+1}(0,T;H^1(I))} + \| q_t \|_{H^{l+1}(0,T;L^2(I))} + \| u \|_{H^{l+1}(0,T;L^2(I))}]\},
\end{aligned}
\tag{64}
$$

*and*

$$
\begin{aligned}
\| u - u^{hk} \|_{L^2(0,T;L^2(I))} &\leq C\{h^{m+1}[\| q \|_{H^1(0,T;H^{m+1}(I))} + \| u \|_{L^2(0,T;H^{m+1}(I))}] \\
&+ k^{l+1}[\| q \|_{H^{l+1}(0,T;H^1(I))} + \| q_t \|_{H^{l+1}(0,T;L^2(I))} + \| u \|_{H^{l+1}(0,T;L^2(I))}]\}.
\end{aligned}
\tag{65}
$$

## 4. Numerical Experiments

Consider the initial boundary value problem of the semilinear convection–diffusion–reaction equation

$$
\begin{cases}
u_t - \epsilon u_{xx} + (2 - x^2)u_x + (1 + xt)\sin u = f(x,t), & x \in I = [0,1], \ t \in J = (0,1], \\
u(0,t) = u_t(0,t) = u(1,t) = u_t(1,t) = 0, & t \in \bar{J}, \quad (66) \\
u(x,0) = u^0(x), & x \in I,
\end{cases}
$$

where the diffusion coefficient $\epsilon$ is positive and the source term $f(x,t) = e^{-t}(c_1 + c_2 x - c_3) + (1 - e^{-t})\frac{c_3}{\epsilon} + (2 - x^2)(1 - e^{-t})(c_2 - \frac{c_3}{\epsilon}) + (1 + xt)\sin[(1 - e^{-t})(c_1 + c_2 x - c_3)]$. The exact solution is $u(x,t) = (1 - e^{-t})(c_1 + c_2 x - c_3)$ and $q = (1 - e^{-t})(c_2 - \frac{c_3}{\epsilon})$, where $c_1 = e^{\frac{-1}{\epsilon}}, c_2 = 1 - c_1, c_3 = e^{\frac{-1+x}{\epsilon}}$.

The proposed method in this work combines time and space variables, so the 1D problem can be viewed as a 2D problem. Here, the space-time domain $[0,1] \times [0,1]$ is partitioned into $m \times n$ rectangular elements. The space-time linear and quadratic polynomial basis functions are taken in this experiment. Here, the convergence orders are calculated by using the following formula

$$
order = \frac{\log[a_i/a_{i+1}]}{\log[\delta_i/\delta_{i+1}]}, \ i = 1,2,3,4,5,
$$

where $a_i$ is the error and $\delta_i$ is the step size.

First, we consider the order of convergence in space direction. For this purpose, a sufficiently small fixed time step $k$ is fixed (ensuring that the time part of the error is a very small percentage of the overall error), and the space grid parameter $h$ is reduced by half. Table 1 gives the errors and convergence orders of $q - q^{hk}$ and $u - u^{hk}$ in the $L^2([0,T]; L^2(I))$ norm for the linear polynomial basis functions with a fixed time step $k = 1/500$, respectively. It can be seen from Table 1 that the convergence orders of $q - q^{hk}$ and $u - u^{hk}$ are close to the second-order under the $L^2([0,T]; L^2(I))$ norm. Similarly, the errors and space convergence orders for a quadratic basis function with the fixed time step $k = 1/200$ are presented in Table 2. Nearly third-order convergence rates can be seen from Table 2.

**Table 1.** Error and order of convergence in space direction with linear basis function.

| | h | $\|\mathbf{q} - \mathbf{q^{hk}}\|_{\mathbf{L^2([0,T];L^2(I))}}$ | Order | $\|\mathbf{u} - \mathbf{u^{hk}}\|_{\mathbf{L^2([0,T];L^2(I))}}$ | Order |
|---|---|---|---|---|---|
| | | | **k = 1/500** | | |
| $\epsilon = 1$ | 1/2 | $5.0612 \times 10^{-3}$ | – | $5.7799 \times 10^{-3}$ | – |
| | 1/4 | $1.2387 \times 10^{-3}$ | 2.0307 | $1.4489 \times 10^{-3}$ | 1.9961 |
| | 1/8 | $3.0813 \times 10^{-4}$ | 2.0072 | $3.6247 \times 10^{-4}$ | 1.9990 |
| | 1/16 | $7.6934 \times 10^{-5}$ | 2.0018 | $9.0638 \times 10^{-5}$ | 1.9997 |
| | 1/32 | $1.9224 \times 10^{-5}$ | 2.0007 | $2.2666 \times 10^{-5}$ | 1.9996 |
| $\epsilon = 0.1$ | 1/2 | 1.3167 | – | $1.5796 \times 10^{-1}$ | – |
| | 1/4 | $3.4350 \times 10^{-1}$ | 1.9385 | $3.4259 \times 10^{-2}$ | 2.2050 |
| | 1/8 | $9.1683 \times 10^{-2}$ | 1.9056 | $9.5018 \times 10^{-3}$ | 1.8502 |
| | 1/16 | $2.3474 \times 10^{-2}$ | 1.9656 | $2.4611 \times 10^{-3}$ | 1.9489 |
| | 1/32 | $5.9090 \times 10^{-3}$ | 1.9901 | $6.2080 \times 10^{-4}$ | 1.9871 |

**Table 2.** $\epsilon = 1$, error and order of convergence in space direction with quadratic basis function.

| h | $\|\mathbf{q} - \mathbf{q^{hk}}\|_{\mathbf{L^2([0,T];L^2(I))}}$ | Order | $\|\mathbf{u} - \mathbf{u^{hk}}\|_{\mathbf{L^2([0,T];L^2(I))}}$ | Order |
|---|---|---|---|---|
| | | **k = 1/200** | | |
| 1/2 | $1.9965 \times 10^{-4}$ | – | $1.9186 \times 10^{-4}$ | – |
| 1/4 | $2.4386 \times 10^{-5}$ | 3.0334 | $2.4156 \times 10^{-5}$ | 2.9896 |
| 1/8 | $3.0327 \times 10^{-6}$ | 3.0074 | $3.0256 \times 10^{-6}$ | 2.9971 |
| 1/16 | $3.7863 \times 10^{-7}$ | 3.0018 | $3.7840 \times 10^{-7}$ | 2.9992 |
| 1/32 | $4.7362 \times 10^{-8}$ | 2.9990 | $4.7310 \times 10^{-8}$ | 2.9997 |

Next, we consider the order of convergence in time direction. Hence, a small fixed space step $h$ is taken, and then the time step $k$ is decreased in a certain proportion. Table 3 gives the errors and convergence orders of $q - q^{hk}$ and $u - u^{hk}$ in the $L^2([0, T]; L^2(I))$ norm for the linear polynomial basis functions with a fixed time step $h = 1/500$, respectively. It can be seen from Table 3 that the convergence orders of $q - q^{hk}$ and $u - u^{hk}$ are also close to the second-order under the $L^2([0, T]; L^2(I))$ norm. Similarly, the errors and the convergence orders for a quadratic basis function with a fixed space step $h = 1/1000$ are given in Table 4. Nearly third-order convergence rates can also be seen in Table 4. These numerical results are consistent with the theoretical analysis results of Theorem 4.

**Table 3.** Error and order of convergence in temporal direction with linear basis function.

| | k | $\|\mathbf{q} - \mathbf{q^{hk}}\|_{\mathbf{L^2([0,T];L^2(I))}}$ | Order | $\|\mathbf{u} - \mathbf{u^{hk}}\|_{\mathbf{L^2([0,T];L^2(I))}}$ | Order |
|---|---|---|---|---|---|
| | | | **h = 1/500** | | |
| $\epsilon = 1$ | 1/2 | $2.7259 \times 10^{-3}$ | – | $8.4522 \times 10^{-4}$ | – |
| | 1/4 | $6.9177 \times 10^{-4}$ | 1.9784 | $2.1407 \times 10^{-4}$ | 1.9812 |
| | 1/8 | $1.7365 \times 10^{-4}$ | 1.9941 | $5.3720 \times 10^{-5}$ | 1.9946 |
| | 1/16 | $4.3440 \times 10^{-5}$ | 1.9990 | $1.3472 \times 10^{-5}$ | 1.9954 |
| | 1/32 | $1.0853 \times 10^{-5}$ | 2.0010 | $3.4049 \times 10^{-6}$ | 1.9843 |
| $\epsilon = 0.1$ | 1/2 | $2.9338 \times 10^{-2}$ | – | $6.2969 \times 10^{-3}$ | – |
| | 1/4 | $7.4294 \times 10^{-3}$ | 1.9815 | $1.5764 \times 10^{-3}$ | 1.9980 |
| | 1/8 | $1.8565 \times 10^{-3}$ | 2.0007 | $3.9322 \times 10^{-4}$ | 2.0032 |
| | 1/16 | $4.5696 \times 10^{-4}$ | 2.0225 | $9.8075 \times 10^{-5}$ | 2.0034 |
| | 1/32 | $1.0822 \times 10^{-4}$ | 2.0780 | $2.4453 \times 10^{-5}$ | 2.0039 |

**Table 4.** $\epsilon = 1$, error and order of convergence in temporal direction with quadratic basis function.

| | | **h = 1/1000** | | |
| k | $\|q - q^{hk}\|_{L^2([0,T];L^2(I))}$ | Order | $\|u - u^{hk}\|_{L^2([0,T];L^2(I))}$ | Order |
|---|---|---|---|---|
| 1/2 | $8.9522 \times 10^{-5}$ | – | $2.7813 \times 10^{-5}$ | – |
| 1/3 | $2.7606 \times 10^{-5}$ | 2.9015 | $8.4885 \times 10^{-6}$ | 2.9270 |
| 1/4 | $1.2180 \times 10^{-5}$ | 2.8441 | $3.6941 \times 10^{-6}$ | 2.8920 |
| 1/5 | $6.5664 \times 10^{-6}$ | 2.7689 | $1.9607 \times 10^{-6}$ | 2.8387 |
| 1/6 | $4.0216 \times 10^{-6}$ | 2.6891 | $1.1817 \times 10^{-6}$ | 2.7772 |

Further, Table 5 gives the errors and convergence orders of $q - q^{hk}$ and $u - u^{hk}$ in the $L^2([0,T]; L^2(I))$ norm for the quadratic polynomial basis functions with the same mesh size $h = k$, respectively. Nearly third-order convergence rates can also be seen from Table 5.

**Table 5.** When h = k, errors and orders of convergence with quadratic basis function.

| | h = k | $\|q - q^{hk}\|_{L^2([0,T];L^2(I))}$ | Order | $\|u - u^{hk}\|_{L^2([0,T];L^2(I))}$ | Order |
|---|---|---|---|---|---|
| | 1/8 | $2.7995 \times 10^{-5}$ | – | $2.5708 \times 10^{-5}$ | – |
| $\epsilon = 1$ | 1/10 | $2.5708 \times 10^{-5}$ | 3.0139 | $1.2968 \times 10^{-5}$ | 3.0667 |
| | 1/20 | $3.5971 \times 10^{-6}$ | 2.9348 | $3.1283 \times 10^{-6}$ | 3.0256 |
| | 1/25 | $1.9081 \times 10^{-6}$ | 2.8412 | $1.6018 \times 10^{-6}$ | 2.9998 |
| | 1/8 | $9.2156 \times 10^{-3}$ | – | $9.2923 \times 10^{-4}$ | – |
| $\epsilon = 0.1$ | 1/10 | $4.9007 \times 10^{-3}$ | 2.8301 | $4.9290 \times 10^{-4}$ | 2.8415 |
| | 1/20 | $6.4658 \times 10^{-4}$ | 2.9221 | $6.4782 \times 10^{-5}$ | 2.9276 |
| | 1/25 | $3.3331 \times 10^{-4}$ | 2.9695 | $3.3380 \times 10^{-5}$ | 2.9715 |

Figure 1 shows the comparison between the numerical solutions $u^{hk}, q^{hk}$ and the exact solutions $u, q$ with step sizes $h = \frac{1}{80}, k = \frac{1}{80}$ under the space-time linear basis functions for $\epsilon = 1.0, \epsilon = 0.1$, and $\epsilon = 10^{-2}$, respectively. From the figure, the numerical solution is a good simulation of the exact solution.

When the diffusion parameter $\epsilon$ is sufficiently small, however, the scheme will be unstable. That is, there will be oscillations produced in areas with large gradient changes. Thus, we need to introduce a stabilizing term in the scheme to deal with this phenomenon. For example, we can use the local projection stabilization technique [7,32].

Furthermore, we compare the proposed method with the traditional $H^1$-Galerkin mixed finite element method combined with the Crank–Nicolson time difference discretization from the perspective of the time direction convergence orders and errors at the final time $t = T$. The numerical results for the piecewise linear polynomial basis functions are given in Tables 6 and 7. These numerical data show that the errors of the proposed method $\|u - u^{hk}\|_{L^2(I)}, \|q - q^{hk}\|_{L^2(I)}$ and that of the traditional method $\|u - u^{CN}\|_{L^2(I)}, \|q - q^{CN}\|_{L^2(I)}$ are almost identical and present an almost second-order convergence order in the time direction.

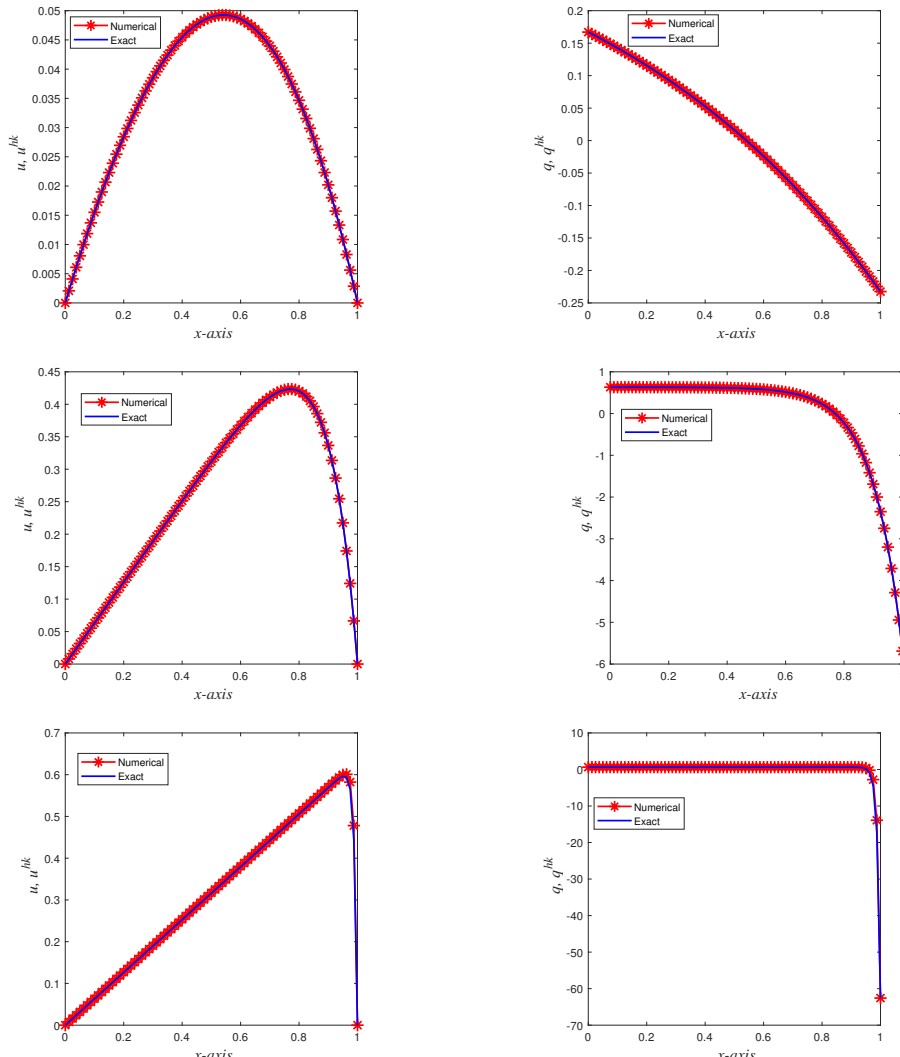

**Figure 1.** Take t = T, left: the exact and numerical solutions of u, right: the exact and numerical solutions of q; top to bottom: $\epsilon = 1.0$, $\epsilon = 0.1$ and $\epsilon = 10^{-2}$.

**Table 6.** Linear basis function. Take t = T.

|  | k(h = 2k) | $\|u - u^{CN}\|_{L^2(I)}$ | Order | $\|u - u^{hk}\|_{L^2(I)}$ | Order |
|---|---|---|---|---|---|
| | 1/25 | $1.4615 \times 10^{-5}$ | — | $1.5646 \times 10^{-5}$ | — |
| | 1/35 | $7.4626 \times 10^{-6}$ | 1.9976 | $8.0583 \times 10^{-6}$ | 1.9720 |
| $\epsilon = 1$ | 1/45 | $4.5157 \times 10^{-6}$ | 1.9988 | $4.9038 \times 10^{-6}$ | 1.9764 |
| | 1/55 | $3.0233 \times 10^{-6}$ | 1.9993 | $3.2964 \times 10^{-6}$ | 1.9793 |
| | 1/65 | $2.1648 \times 10^{-6}$ | 1.9996 | $2.3675 \times 10^{-6}$ | 1.9812 |
| | 1/25 | $3.9251 \times 10^{-4}$ | — | $3.8624 \times 10^{-4}$ | — |
| | 1/35 | $2.0038 \times 10^{-4}$ | 1.9982 | $1.9717 \times 10^{-4}$ | 1.9983 |
| $\epsilon = 0.1$ | 1/45 | $1.2125 \times 10^{-4}$ | 1.9990 | $1.1931 \times 10^{-4}$ | 1.9989 |
| | 1/55 | $8.1177 \times 10^{-5}$ | 1.9994 | $7.9882 \times 10^{-5}$ | 1.9992 |
| | 1/65 | $5.8125 \times 10^{-5}$ | 1.9996 | $5.7200 \times 10^{-5}$ | 1.9993 |

**Table 7.** Linear basis function. Take t = T.

| | k(h = 2k) | $\|q - q^{CN}\|_{L^2(I)}$ | Order | $\|q - q^{hk}\|_{L^2(I)}$ | Order |
|---|---|---|---|---|---|
| | 1/25 | $1.5752 \times 10^{-5}$ | – | $1.3716 \times 10^{-5}$ | – |
| | 1/35 | $8.0203 \times 10^{-6}$ | 2.0061 | $7.0257 \times 10^{-6}$ | 1.9883 |
| $\epsilon = 1$ | 1/45 | $4.8472 \times 10^{-6}$ | 2.0038 | $4.2607 \times 10^{-6}$ | 1.9764 |
| | 1/55 | $3.2431 \times 10^{-6}$ | 2.0027 | $2.8570 \times 10^{-6}$ | 1.9917 |
| | 1/65 | $2.3212 \times 10^{-6}$ | 2.0021 | $2.0480 \times 10^{-6}$ | 1.9928 |
| | 1/25 | $3.7154 \times 10^{-3}$ | – | $3.7156 \times 10^{-3}$ | – |
| | 1/35 | $1.8964 \times 10^{-3}$ | 1.9988 | $1.8946 \times 10^{-3}$ | 2.0017 |
| $\epsilon = 0.1$ | 1/45 | $1.1474 \times 10^{-3}$ | 1.9993 | $1.1455 \times 10^{-3}$ | 2.0023 |
| | 1/55 | $7.6816 \times 10^{-4}$ | 1.9996 | $7.6642 \times 10^{-4}$ | 2.0025 |
| | 1/65 | $5.5001 \times 10^{-4}$ | 1.9997 | $5.4851 \times 10^{-4}$ | 2.0025 |

However, one can observe from the data obtained by using the piecewise quadratic polynomial basis functions in Tables 8 and 9 that the errors of the proposed method are almost one order of magnitude smaller than that of the traditional method. Moreover, the time direction convergence order of the proposed method is close to third-order, while that of the traditional method is still close to second-order since it uses the Crank–Nicolson scheme in the time direction. This implies that the proposed method in this paper can improve the convergence order and calculation accuracy by increasing the polynomial degree of the basis functions and allowing large time steps. Therefore, the space-time mixed $H^1$-Galerkin scheme proposed in this paper is superior to the traditional mixed $H^1$-Galerkin scheme.

**Table 8.** Quadratic basis function. Take t = T.

| | k(h = 2k) | $\|u - u^{CN}\|_{L^2(I)}$ | Order | $\|u - u^{hk}\|_{L^2(I)}$ | Order |
|---|---|---|---|---|---|
| | 1/25 | $5.1157 \times 10^{-7}$ | – | $1.9517 \times 10^{-8}$ | – |
| | 1/35 | $2.6450 \times 10^{-7}$ | 1.9605 | $7.0665 \times 10^{-9}$ | 3.0193 |
| $\epsilon = 1$ | 1/45 | $1.6071 \times 10^{-7}$ | 1.9826 | $3.3135 \times 10^{-9}$ | 3.0137 |
| | 1/55 | $1.0777 \times 10^{-7}$ | 1.9914 | $1.8110 \times 10^{-9}$ | 3.0106 |
| | 1/65 | $7.7214 \times 10^{-8}$ | 1.9957 | $1.0956 \times 10^{-9}$ | 3.0085 |
| | 1/25 | $2.5049 \times 10^{-5}$ | – | $6.4858 \times 10^{-6}$ | – |
| | 1/35 | $1.2537 \times 10^{-5}$ | 2.0571 | $2.3669 \times 10^{-6}$ | 2.9960 |
| $\epsilon = 0.1$ | 1/45 | $7.5220 \times 10^{-6}$ | 2.0327 | $1.1143 \times 10^{-6}$ | 2.9976 |
| | 1/55 | $5.0140 \times 10^{-6}$ | 2.0212 | $6.1051 \times 10^{-7}$ | 2.9983 |
| | 1/65 | $3.5810 \times 10^{-6}$ | 2.0149 | $3.6995 \times 10^{-7}$ | 2.9986 |

**Table 9.** Quadratic basis function. Take t = T.

| | k(h = 2k) | $\|q - q^{CN}\|_{L^2(I)}$ | Order | $\|q - q^{hk}\|_{L^2(I)}$ | Order |
|---|---|---|---|---|---|
| | 1/25 | $5.2298 \times 10^{-6}$ | – | $2.1434 \times 10^{-8}$ | – |
| | 1/35 | $2.6555 \times 10^{-6}$ | 2.0143 | $7.6875 \times 10^{-9}$ | 3.0475 |
| $\epsilon = 1$ | 1/45 | $1.6024 \times 10^{-6}$ | 2.0099 | $3.5791 \times 10^{-9}$ | 3.0419 |
| | 1/55 | $1.0711 \times 10^{-6}$ | 2.0076 | $1.9452 \times 10^{-9}$ | 3.0384 |
| | 1/65 | $7.6607 \times 10^{-7}$ | 2.0062 | $1.1715 \times 10^{-9}$ | 3.0356 |
| | 1/25 | $1.9216 \times 10^{-4}$ | – | $6.4817 \times 10^{-5}$ | – |
| | 1/35 | $9.5103 \times 10^{-5}$ | 2.0905 | $2.3657 \times 10^{-5}$ | 2.9955 |
| $\epsilon = 0.1$ | 1/45 | $5.6760 \times 10^{-5}$ | 2.0537 | $1.1138 \times 10^{-5}$ | 2.9974 |
| | 1/55 | $3.7725 \times 10^{-5}$ | 2.0357 | $6.1025 \times 10^{-6}$ | 2.9983 |
| | 1/65 | $2.6895 \times 10^{-5}$ | 2.0256 | $3.6978 \times 10^{-6}$ | 2.9988 |

## 5. Conclusions

By introducing the auxiliary variable $q = a(x)u_x$, we first obtain a coupled system equivalent to problem (1). The $H^1$-Galerkin space-time mixed finite element method is established for the coupled system (4). The finite element discrete is utilized in both space and time directions. Therefore, the space-time mixed finite element approach can concurrently obtain formal higher-order precision of the space and time variables. The uniqueness of the approximate solutions $u$ and $q$ are demonstrated. The $L^2(L^2)$ norm estimates of the approximate solution $u$ and $q$ are proven by introducing the space-time projection operator without the constraint of space-time grid conditions. Finally, we give the numerical simulation of the original problem. Numerical experiments verify the correctness of the analysis. Furthermore, by comparing with the classical H1-Galerkin mixed finite element scheme, the proposed scheme can easily improve computational accuracy and time convergence order by changing the basis function.

Significantly, from the above analysis, it can be seen that the $H^1$-Galerkin space-time mixed finite element method is discrete by finite element in both time and space directions, so it will generate more degrees of freedom than the standard finite element method, which will greatly affect the efficiency of the algorithm. Therefore, the proper orthogonal decomposition technology to reduce its dimension is the subject of our follow-up research.

**Author Contributions:** Conceptualization, X.R. and H.L.; methodology, X.R. and S.H.; Software, X.R. and S.H.; formal analysis, X.R.; Writing—original draft preparation, X.R.; Writing—review and editing, S.H. and H.L.; supervision, H.L. All authors have read and agreed to the published version of the manuscript.

**Funding:** This research was funded by the National Natural Science Foundation of China (12161063, 12161034), Program for Innovative Research Team in Universities of Inner Mongolia Autonomous Region (NMGIRT2207).

**Data Availability Statement:** Not applicable.

**Acknowledgments:** The authors would like to thank the reviewers and editors for their invaluable comments that greatly refined the content of this article.

**Conflicts of Interest:** The authors declare no conflict of interest.

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
