# Peer review of "An H1-Galerkin Space-Time Mixed Finite Element Method for Semilinear Convection–Diffusion–Reaction Equations"

_fractalfract, doi:10.3390/fractalfract7100757_

Round 1
Reviewer 1 Report
The report is in the attached file.

Reviewer 2 Report
The paper is well-written and presented, results are promising.
Comments to be fixed:
Fix the title Eequations--Equations
The author should mention the Ladyzhenskaya–Babuška–Brezzi condition and then use the abbreviation LBB.
Fix equation (1) t \in J,
In line (39), after method [9], the author should add "space–time ultra-weak discontinuous Galerkin method [Bac2021]". Reference Bac2021 is as follows: M. Baccouch, H. Temimi, A high-order space-time ultra-weak discontinuous Galerkin method for the second-order wave equation in one space dimension, Journal of Computational and Applied Mathematics, Vol. 389, pp. 113331, 2021.
Why did you introduce an auxiliary variable q? Do the error analysis steps change and make it more complicated without q?
How did you treat the nonlinearity in your algorithm?
in Tables 1,2,3,4, why you didn't choose more refined meshes h and k (1/20, 1/30, ... 1/50).
What if you choose the same mesh size in space and time and the same order, quadratic for example, would you have the same rate of convergence 3.
Mention in Figure1 that these graphs are plotted for t=T (I believe)
would you have the optimal rate of convergence if you choose epsilon=0.0001?
In all your simulations, you chose linear or quadratic basis functions, could you go to cubic and above?
A general polish of the English language should be done.
